# LPS2336, a New TREK-1 Channel Activator Identified by High Throughput Screening

**DOI:** 10.3390/biom15050740

**Published:** 2025-05-20

**Authors:** Romane Boyer, Romane Bony, Maxence Maugis, Julien Schopp, Marion Leroux, Clément Michelin, Laurie Gonthier, Quentin Grzeskiewicz, Alexandre Jouannet, Youssef Aissouni, Bruno Didier, Mihaela Gulea, Nicolas Girard, Jean-Christophe Cintrat, Antoine Dumeige, Jérôme Busserolles, Sylvie Ducki, Stéphane Lolignier

**Affiliations:** 1Université Clermont Auvergne, Inserm, CHU Clermont-Ferrand, Neuro-Dol, 63000 Clermont-Ferrand, Francejulien.schopp@uca.fr (J.S.); jerome.busserolles@uca.fr (J.B.); 2Université Clermont Auvergne, CNRS, Clermont Auvergne INP, ICCF, 63000 Clermont-Ferrand, France; 3Université de Strasbourg, CNRS, Laboratoire d’Innovation Thérapeutique, LIT UMR 7200, 67000 Strasbourg, France; bdidier@unistra.fr (B.D.); gulea@unistra.fr (M.G.); nicolas.girard@unistra.fr (N.G.); 4Université Paris-Saclay, CEA, INRAE, Département Médicaments et Technologies pour la Santé (DMTS), SCBM, 91191 Gif-sur-Yvette, France; 5Institut Universitaire de France (IUF), 75231 Paris, France

**Keywords:** TREK-1, high throughput screening, structure–activity relationship, pain, analgesia

## Abstract

TWIK-related K+ (TREK-1) channels are involved in pain perception and their pharmacological activation has potential for pain relief. The development of new pharmacological tools to study these channels and enrich our knowledge of structure–activity relationships is therefore important. We optimized a high throughput screening method based on thallium flux monitoring for the detection of TREK-1 activators in chemical libraries. We screened 1040 compounds from the French National Essential Chemical Library and identified LPS2336 as a potent TREK-1 activator with an EC_50_ of 11.76 µM. Thirty-three LPS2336 analogs were subsequently tested but none of them retained activity on TREK-1. In vivo, LPS2336 produces antinociceptive activity when administered systemically and, to a lesser extent, intracerebroventricularly, but not intrathecally, showing that targeting peripheral TREK-1 channels may be important to produce pain relief, with the interest of reducing potential central adverse effects. LPS2336 was shown to produce sedation and hypothermia with a narrow therapeutic window. As these adverse effects are also observed in TREK-1 knock-out mice, they are likely mediated by off-targets. Our work provides key optimization steps for thallium-based assays and a new pharmacological tool for the study of TREK-1 channels. It also raises the importance of investigating adverse effects in vivo at early stages of drug discovery.

## 1. Introduction

TREK-1 is a member of the two-pore-domain (K2P) potassium channels family. Like other K2P channels, TREK-1 is responsible for a leak (or background) hyperpolarizing current which contributes to the resting membrane potential and opposes membrane depolarization. It is expressed widely in the central and peripheral nervous system, notably in neurons and structures belonging to the pain pathway as well as in many peripheral tissues including lungs, the heart, intestine, ovaries or uterus [1,2,3,4,5,6,7,8]. Accordingly, a role of TREK-1 in different pathophysiological functions has been shown, such as pain [7,9,10,11], depression [12], anesthesia and neuroprotection [13,14,15], cardiac function [16,17], lung injury [18], uterine contraction [19], urination [20] or cancer [21,22]. Regarding pain perception, peripheral TREK-1 channels were shown to be important for the transduction of thermal and mechanical stimuli [3,9,23], and it is not yet clear if or how central TREK-1 channels contribute to pain perception. Interestingly, we have previously observed that TREK-1 channels are important for the analgesic effect of morphine and fentanyl but not in opioidergic adverse effects such as respiratory depression, constipation and dependance [24]. Direct pharmacological activation of TREK-1 might therefore offer a safer analgesic alternative to opioids.

Ourselves and others have previously reported that TREK-1 could be an interesting target to treat pain, with promising in vivo pharmacological proofs-of-concept [11,25,26,27,28]. Our previous work focused on RNE28, an analog of caffeate ester [29] which exhibits high specificity for TREK-1 channels and possesses analgesic properties without inducing opioidergic adverse effects [26]. However, RNE28 has a limited affinity for its target (EC_50_ = 37.4 µM) and further work is needed to identify higher affinity compounds which could be used as tools to investigate the analgesic potential of TREK-1 activation and as a starting point for drug development.

We aimed to optimize a high throughput screening (HTS) approach for the discovery of new TREK-1 activators, based on thallium flux measurement, thanks to the permeability of potassium channels to Tl^+^ ions. This method was previously used to identify activators of other Kir and K2P channels [30,31,32,33]. We have identified important optimization parameters to consider when designing such an assay. One important factor to consider for the screening of K2P channels activators is their leaky nature, allowing potassium efflux at resting membrane potential. In the context of an assay based on the monitoring of thallium influx through K2P channels, activators only accelerate an already rapid thallium intake, and fine tuning the time-point for thallium uptake measurement is key. We also assessed the sensitivity of this assay to DMSO, the solvent of choice for most chemical collections, and found that DMSO concentration is a strong limiting factor of this assay.

Within a collection of 1040 compounds from the French National Essential Chemical Library, we identified two new TREK-1 activators: LPS2336 and RN-1-025. The most potent compound, LPS2336, is an intermediate towards the synthesis of pyridazine derivatives as acetylcholinesterase inhibitors. We observed a reduction in pain thresholds of mice injected with LPS2336 systemically or (to a much lesser extent) centrally into the brain, but not via intrathecal route, showing the importance of biodistribution and the role of peripheral TREK-1 channels in pain. LPS2336 has a narrow therapeutic window as a sedative effect and hypothermia occurs at doses close to the analgesic doses. These effects, however, are not mediated by TREK-1 as they are similarly observed in TREK-1 knock-out (KO) mice. With this preclinical study, we emphasize the importance of conducting early adverse effect evaluation. We propose a set of simple tests requiring limited equipment and training which could help identify subtle adverse reactions in mice that could otherwise be unnoticed.

## 2. Materials and Methods

### 2.1. Animals and Models

Experiments were conducted on 8–12 weeks old male and female CD1 and C57Bl/6J mice. CD1 mice were purchased from Janvier Labs (Le Genest-Saint-Isle, France) and kept under standard conditions (21–22 °C, 12/12 h light/dark cycle, 55% humidity). Upon arrival, animals were given a week to acclimatize before any experimentation. Treatment groups were randomized in equal blocks of 8 animals (a sample size that allows the detection of effect sizes from 0.5 given the variance of our measurements with an alpha risk of 0.05 and power of 80%) and experimenters were blind to the treatment and genotype.

TREK-1^−/−^ mice were generated by crossing mice carrying an allele of the *Kcnk2* gene in which exon 3 was floxed [12] (RRID:MGI:3050295) with ZP3-Cre mice [34] (RRID:MGI:3835429), both in C57Bl6/J background. F1 females, producing TREK-1^+/−^ oocytes, were crossed with WT males. TREK-1^+/−^ F2 offspring not carrying the ZP3-Cre transgene were finally crossed together to generate TREK-1^−/−^ and TREK-1^+/+^ littermates, which were weaned in separate cages after genotyping using 2 mm tail samples taken at 10 days of age. Animals were identified by digital tattoos at 3 weeks of age.

Paw inflammation was induced by intraplantar (i.pl.) injection of 20 µL 2% carrageenan 3 h 30 before administration of LPS2336, morphine or vehicle. LPS2336 is administrated intraperitoneally (i.p.) at a 10 mL/kg volume, intracerebroventricularly (i.c.v.) in 2 µL or intrathecally (i.t.) in 5 µL. LPS2336 and morphine chlorhydrate (Coopération Pharmaceutique Française, Melun, France) were dissolved in 0.9% NaCl on the day of use. λ-carrageenan (Sigma-Aldrich, Saint-Louis, MO, USA) was added to 0.9% NaCl the day prior to the experiment and left to stir overnight at room temperature.

### 2.2. Chemical Synthesis

ML335 and ML67-33 were prepared as previously described by the group of Daniel L Minor [35,36]. The synthesis of LPS2336 and its analogs is described in the Appendix A.

### 2.3. Cell Lines

HEK-293 cells (RRID:CVCL_0045, ATCC# CRL-1573) were grown in DMEM (Gibco, Life Technologies, Carlsbad, CA, USA) supplemented with 10% heat-inactivated fetal calf serum, Glutamine 2 mM, penicillin-streptomycin 100 UI/mL, Vitamins (PAA laboratories # N11-002) at 1X, sodium pyruvate 100 mM, non-essentials amino acids (HiClone # SH30238.01) at 1X in a humidified incubator at 37 °C, 5% CO_2_. For the stable hTREK-1 and hTREK2 cell lines generation, HEK-293 cells (1.5 × 10^5^) were seeded in 6-well plates and transfected with the human *Kcnk2* (NM_001017424) or *Kcnk10* (NM_021161) ORF cloned in a pEZ-M02 expression vector (GeneCopoeia, Rockville, MD, USA, cat #EX-Z2617-M02 and #EX-T0357-M02, respectively) using Jet Prime transfection reagent (Polyplus, Illkirch-Graffenstaden, France) according to the manufacturer’s instructions. Transfected cells were grown in a non-selective medium for 48 h before to be exposed to 800 μg/mL Geneticin (G418, Gibco) for 21 days. Individual G418-resistant cells were then randomly picked and plated into different wells of a 96-well plate and cultured for 2 weeks with 800 μg/mL Geneticin. Resistant clones were considered stable, expanded and cryoconserved after TREK-1 current was confirmed using a patch clamp. Transgenic cells were selected with the addition of 1 mg/mL G418 every 3 passages.

### 2.4. Thallium Assay

Cells were plated in 384-well plates with black walls and clear bottom coated with poly-D-lysine (Greiner Bio-One, Kremsmünster, Austria, #781946) at a density of 25 000 cells/well from a 1 250 000 cells/mL solution. Cell counting was performed with an automatic cell counter (Countness II FL, ThermoFisher, Waltham, MA, USA) with addition of trypan blue 0.2% to exclude dead cells. 24 h after plating, cells were incubated with FluxOR II green probe (ThermoFisher) for 1 h in the dark, accordingly to the supplier’s protocol (‘no wash’ procedure) with addition of 77 mg/mL probenecide. Compounds, diluted in FluxOR II assay buffer with DMSO, were then added to the wells (0.5% DMSO final) for 26 min before the plate was inserted in a Flexstation 3 spectrophotometer (Molecular Devices, San Jose, CA, USA). Fluorescence was measured every 3 s during 1 min (excitation 485 nm, emission 515 nm), and 0.5 mM thallium was added to the wells after a 20 s baseline. Note that, for the characterization of LPS2336 analogs, performed at a different time on a different batch of HEK-TREK-1 cells, the thallium concentration had to be raised to 2 mM, likely due to a lower TREK-1 expression in these cells. Fluorescence is normalized using the formula (F − F0)/F0 where F0 is the average fluorescence during the first 18 s of the recording. ΔFmax/F0 is calculated between 18 and 33 s (−2 to 13 s relative to thallium addition). For dose–response curves, ΔFmax/F0 are expressed as %vehicle (100% corresponding to the average ΔFmax/F0 measured in vehicle-treated wells). For high throughput screening, Z scores were calculated as(1)$$Z=\frac{X_{sample}-{\bar{X}}_{neg}}{\sigma_{neg}}$$
where X represents ΔFmax/F0 and σ the ΔFmax/F0 standard deviation. ‘neg’ represents negative controls (vehicle-treated wells).

Different quality controls were performed on each HTS plate [37]. Signal-to-background was calculated as(2)$$SB=\frac{{\bar{Z}}_{pos}}{{\bar{Z}}_{neg}}$$
where Z_pos_ and Z_neg_ represent the Z score of positive controls (25 µM BL-1249) and negative controls, respectively. Signal-to-noise was calculated as(3)$$SN=\frac{{\bar{Z}}_{pos}-{\bar{Z}}_{neg}}{\sqrt{\sigma_{pos}^{2}+\sigma_{neg}^{2}}}$$

Z’ factor was calculated as(4)$$Z\prime =1-\frac{3\sigma_{pos}+3\sigma_{neg}}{\left|{\bar{Z}}_{pos}-{\bar{Z}}_{neg}\right|}$$

Strictly standardized mean difference (SSMD) was calculated as(5)$$SSMD=\frac{{\bar{Z}}_{pos}+{\bar{Z}}_{neg}}{\sqrt{\sigma_{pos}^{2}+\sigma_{neg}^{2}}}$$

The screening was performed on a collection of 1040 compounds supplied by the French Chemical Library (ChemBioFrance, Essential Chemical Library).

### 2.5. Patch Clamp

Cells are plated on 35 mm dishes at a density of 30,000 cells per dish and whole cell patch clamp recordings were performed 24 to 48 h after plating. Culture medium was replaced by 1 mL extracellular solution containing (in mM): NaCl 140, tetrathylamonium-Cl 10, KCl 5, MgCl_2_ 3, CaCl_2_ 1, HEPES 10. pH was adjusted to 7.4 with NaOH and osmolarity to 305 mOsm with sucrose. Intracellular solution contained (in mM): KCl 155, MgCl_2_ 3, EGTA 5, HEPES 10. pH was adjusted to 7.4 with KOH and osmolarity to 310 mOsm with sucrose. Currents were recorded with an Axopatch 200B amplifier (Axon Instruments, Union City, CA, USA) and digitalized at 10 kHz with a Digidata 1444 (Axon Instruments). Clampex 10.7 software (Axon Instruments) was used for voltage clamping and current recording. Patch pipettes resistance was between 2.5 and 4.0 MΩ. After whole cell configuration was achieved, cells were maintained at a resting membrane potential of −80 mV. Ten −100 to +60 mV voltage ramps were then applied in 40 ms every 5 s. As saturation of the amplifier occurred in some whole cell recordings above +30 mV (this was seen when large TREK-1 conductances were induced by pharmacological activators of the channel), only currents generated below +30 mV were analyzed. Currents were recorded between 5 and 60 min after the addition of intracellular solution in which tested compounds were diluted. Analyses were performed on mean current densities (current divided by the cell capacitance, I/Cm) measured at 0 mV over 10 consecutive stimulations.

### 2.6. Behavioral Tests

For all behavior tests, mice were acclimatized to the experiment room for at least 30 min before the test.

#### 2.6.1. Hargreaves Test

Mice were placed in individual plexiglass boxes (128 × 96 × 96 mm) set on a glass plate heated at 29 °C (Hargreaves analgesimeter, IITC Life Science, Woodland Hills, CA, USA) for habituation during 1 h 30. Pain thresholds were evaluated by measuring the paw withdrawal latency in response to a heat stimulus focally applied to the hindpaw through the glass by an infrared beam. The infrared source was set to 40% of the maximal intensity and a cut-off of 20 s was applied to avoid tissue damage. For the baseline, up to 5 threshold measurements were performed (10 min apart) until two measures less than 1 s apart were obtained and averaged. After administration of the treatments, a time-course evaluation of pain thresholds was performed by conducting one measure at 0, 15, 30, 45, 60, 90 and 120 min.

#### 2.6.2. Open Field

Mice were placed in the center of a 50 × 50 cm arena with 45 cm high walls. Animals were filmed for 10 min and the distance traveled was measured using video-tracking software (Viewpoint Videotrack 5.13.0.10).

#### 2.6.3. Horizontal Bar Test

A 13 cm long, 2 mm diameter metal bar was placed horizontally between 2 rods at a 30 cm height. To evaluate the general wakefulness of animals, and reveal a potential sedative effect of tested compounds, mice were lifted by the tail and placed close to the bar so that they naturally grab it with both forepaws. After gently releasing the tail, the time taken by mice to climb on the bar (grabbing it with their four paws) was monitored.

#### 2.6.4. Rotarod

To evaluate motor coordination, mice were placed on a 9 cm diameter rod with increasing rotating speed until they fell (rotarod apparatus, Bioseb, Vitrolles, France). The day before testing, animals were trained to the test until they could stay on the rod rotating at 4 rpm for 60 s. On test day, animals were placed on the rod at 4 rpm and the apparatus was set to accelerate from 4 to 40 rpm in 300 s. The test began when acceleration started and ended when the animal fell off the rod. The procedure was repeated three times with a 15 min recovery interval and the average latency to fall was calculated for each animal.

### 2.7. Body Temperature Monitoring

Body temperature was measured using a rectal probe in awake animals before and 15, 30 and 45 min after injection of LPS2336.

### 2.8. Data Analysis

Data treatment was performed using Microsoft Excel 365 and Python 3.10 code (for high throughput screening, using the functions detailed in the corresponding method section). Statistical analyzes were performed using GraphPad Prism 10. Group sizes, error calculation, statistical tests and significance thresholds are indicated in the figure legends. Parametric tests were used only when values were normally distributed around the mean in all groups (D’Agostino and Pearson test). No animal was excluded from analyses.

## 3. Results

### 3.1. Optimization of the Thallium Assay

Potassium channels, including K2P channels, are permeant to thallium ions. This makes them suitable targets for high throughput screening (HTS) using fluorescence-based thallium flux assays [30,31]. We used the FluxOR II Green Potassium Ion Channel Assay (Invitrogen, Carlsbad, CA, USA) to search for TREK-1 channel activators in a chemical compound collection. This assay consists of loading TREK-1-expressing HEK293 cells with a membrane-permeant fluorescent thallium indicator before adding test compounds to the wells. Thallium is then added to the culture wells which results in an increase in fluorescence whose rate depends on the permeability of the cell membrane to thallium and follows a logarithmic function. This is due to the progressive diminution of the Tl^+^ ion’s driving force as these ions get closer to equilibrium between intra- and extra-cellular compartments. Because of the non-linearity of the response, we had to define the optimal time-point after thallium injection to acquire fluorescence data together with the optimal thallium concentration. Finally, as the compounds used for high throughput screening were provided in DMSO, we also needed to study the effect of DMSO concentration on our assay. For these optimization steps, we focused on the impact of these parameters on the ability of the assay to discriminate between untreated TREK-1-expressing cells and cells treated with a known TREK-1 activator. We used 10 µM BL-1249, which was the most potent TREK-1 activator available at the time of this work [38].

First, we generated a stable TREK-1-expressing cell line by transfecting the human *Kcnk2* coding sequence in HEK293 cells upstream of a CMV promoter (HEK-hTREK-1). Monoclonal cultures were produced after antibiotic selection of cells and a single clone was selected and amplified after confirmation of a TREK-1 current by patch clamp. To optimize thallium concentration, we carried out a dose–response experiment with final thallium concentrations ranging from 2 µM to 4 mM (Figure 1A). Only a marginal fluorescence increase was observed in naive HEK293 cells compared to HEK-hTREK-1 cells (Figure 1B). BL-1249 treatment had no effect on naive cells but produced a strong increase in fluorescence (normalized as ΔFmax/F0) in HEK-hTREK-1 cells. Normalizing thallium dose–response data from 0 to 100% allowed us to compare the effect of thallium concentration in vehicle- and BL-1249-treated cells, showing a more linear [Tl^+^]/signal relationship in treated cells than in unstimulated cells, in which the [Tl^+^]/signal curve follows a function closer to an exponential (Figure 1C). Thallium concentrations around 0.25–0.5 mM (in the knee of the vehicle curve) seemed to allow the best separation between the two conditions. We calculated the ΔFmax/F0 ratio between treated and non-treated cells which was highest between 0.125 and 0.5 mM (Figure 1D). As the manufacturer of the FluxOR II Green probe recommended a range of 0.5 to 4 mM thallium, we selected 0.5 mM for subsequent experiments and HTS.

Compounds in our chemical collection were provided at 5 mM in 100% DMSO. While they were diluted in a saline buffer for HTS, residual DMSO might be toxic for cells and impact the test. To determine what concentration of compounds could be used in HTS, we performed a DMSO dose–response experiment (Figure 1E). High DMSO concentrations decreased the ΔFmax/F0 signal, especially in untreated HEK-hTREK-1 cells. This is more striking when the signal in treated and untreated cells is normalized from 0 to 100% (Figure 1F). As a good compromise, we chose a final DMSO concentration of 0.5%, which did not affect the signal in BL-1249-treated cells and had only a limited impact in untreated cells (84.5% of residual signal when compared to the lowest DMSO concentration tested). Most importantly, 0.5% final DMSO allowed the HTS to be performed with compounds at 25 µM.

Another important parameter to consider is the time-point at which Fmax is measured. To optimize this parameter, we conducted parallel analyses of the same data (HEK-hTREK-1 cells treated with vehicle or 10 µM BL-1249 in 0.5% DMSO), only varying the Fmax time-point. Again, we calculated the ΔFmax/F0 ratio between BL-1249- and vehicle-treated cells (Figure 1G). Early time-points (1 to 7 s) resulted in a high variability of ratios. From 10 to 26 s after thallium injection, ratios were much more consistent between wells, with a peak at 13 s, offering the best separation between the two conditions. We also calculated Z scores of BL-1249-treated wells (Figure 1H). The 13 s time-point was the one showing the smallest standard deviation, close to the maximum Z score. We therefore selected 13 s for all subsequent analyses and the HTS. Note that all experiments in Figure 1 are shown after reanalysis using this parameter.

BL-1249 is a highly potent TREK-1 activator, and we wanted to test how this assay, using our optimized conditions, performed with other TREK-1 modulators. We selected eight known TREK-1 activators—arachidonic acid [39], flufenamic acid [40], riluzole [41], ML67-33 [35], ML335, ML402 [36], GI-530159 [42], BL-1249—and one inhibitor—fluoxetine [43]. Then, we ran dose–effect experiments (Figure 2). To improve aqueous solubility of ML335, ML402 and GI-530159, salts were also prepared (disodium, potassium, and chlorhydrate salts, respectively). All TREK-1 activators produced TREK-1 activation in the 10–100 µM range, while BL-1249 was active at sub-micromolar concentrations (and induced visible cell death and a drop in signal intensity at 100 µM). Salification resulted in an increased TREK-1 activation at 100 µM for ML335 and ML402. At 100 µM, fluoxetine reduced TREK-1 activation by half (54% of vehicle).

### 3.2. High Throughput Screening for TREK-1 Activators

Using the parameters optimized previously, we screened a collection of 1040 compounds (‘Essential Collection’) from the French National Chemical Library, ChemBioFrance. Compounds were added to wells at a 25 µM final concentration 26 min before thallium addition. Compounds were spread on four 384-well plates, including 15 vehicle-treated wells (negative controls) and 15 BL-1249-treated wells (positive controls, also at 25 µM) on each plate. These plates were run in triplicates. Z scores were calculated relative to vehicle-treated wells within each plate. The average coefficient of variation was 3.05 ± 0.95 for positive controls over the 12 plates, and 17.95 ± 3.02 for negative controls. We calculated different quality control parameters over the assay [37]. Signal-to-background was 12.02 ± 0.68 (min value 11.28). Signal-to-noise was 62.90 ± 10.64 (min value 39.81). Z’ factor was 0.85 ± 0.03 (min value 0.77). Strictly standardized mean difference (SSMD) was 28.12 ± 6.24 (min value 16.13). Finally, in negative control wells, Z scores ranged from −2.96 to +2.84, with a 90% percentile of 1.33 (*n* = 180 over 12 plates). We therefore determined a threshold for hit detection of 3 to avoid false positive detection (Figure 3A).

Applying this threshold, two compounds were identified as hits on at least one replicate (Figure 3B). One compound, later identified as LPS2336, reached the positivity threshold 3/3 times with an average Z score of 3.50 ± 0.24. LPS2336 (Figure 3C) is a benzylated piperidine containing a nitrile function prepared by Camille G Wermuth [44] as an intermediate towards the synthesis of pyridazine derivatives as acetylcholinesterase inhibitors. Another compound, RN-1-025, reached the positivity threshold in one replicate only, with an average Z score of 2.68 ± 0.49. RN-1-025 (Figure 3C) is a disubstituted quinazoline prepared by Jean-Christophe Cintrat [45] as a protective molecule against Shiga toxins.

To evaluate the drug-likeness and pharmacokinetic potential of LPS2336, the most potent of the two compounds according to the screening results, we conducted an in silico evaluation using *molinspiration* (https://www.molinspiration.com/), *SwissADME* (http://www.swissadme.ch/), and the toxicity module of *preADMET* (https://preadmet.webservice.bmdrc.org/toxicity/, accessed on 6 May 2025). The physicochemical properties of LPS2336 are within the range of drug-likeness, with moderate lipophilicity (log P 2.71), low polar surface area (PSA 27.03 Å^2^) and good solubility. LPS2336 complies with Lipinski’s rule of 5. These features support efficient membrane permeability and oral bioavailability, reflected by a bioavailability score of 0.55. The piperidine is predicted to have high gastrointestinal absorption and is blood–brain-barrier permeant, meaning that it should reach the central nervous system (CNS). Importantly, LPS2336 does not inhibit major CYPs involved in drug elimination. From a medicinal chemistry perspective, the lack of pan-assay interference compounds (PAINS) alert and a low synthetic accessibility score (1.39) further supports its suitability for lead optimization. However, predicted toxicity flags LPS2336 as mutagenic in the Ames test (probably because of the cyanide moiety) and exhibits a medium risk for hERG inhibition, suggesting potential genotoxicity and cardiotoxicity, and highlighting the potential need for future structural refinement to improve its safety.

### 3.3. Confirmation of TREK-1 Activation by LPS2336

To validate and further characterize piperidine LPS2336, as well as quinazoline RN-1-025, as TREK-1 activators, we tested their effect on TREK-1 using patch clamp. At 50 µM, LPS2336 increased the mean current density recorded at 0 mV in HEK-hTREK-1 cells by 10.77-fold (Figure 4A). In accordance with our HTS results, RN-1-025 was less potent with a current increase of 6.01-fold. We used ML335 and GI-530159 as references, which induced a current increase of 13.35 and 9.73-fold, respectively. Riluzole did not induce a significant increase in TREK-1 current at this concentration (2.23-fold).

As LPS2336 produced stronger TREK-1 activation than RN-1-025, and since RN-1-025 was poorly soluble in water, making future parenteral administration in mice impossible, we decided to focus on the characterization and optimization of LPS2336. To improve the aqueous solubility of LPS2336 further, hydrochloride salt was prepared (LPS2336.HCl). In HEK-hTREK-1 cells, LPS2336.HCl induced a current increase comparable to that of LPS2336 (Figure 4B). Therefore, LPS2336.HCl replaced LPS2336 in subsequent experiments.

We then generated dose–response curves for LPS2336.HCl using the thallium assay. LPS2336.HCl induced an increase in ΔFmax/F0 of 2.26-fold at the highest concentration tested (100 µM) compared to the lowest concentration tested (1.7 nM). In comparison, ML335, used as a positive control, induced a 2.60-fold increase in ΔFmax/F0 (Figure 4C). EC_50_ were 11.76 µM for LPS2336.HCl and 6.7 µM for ML335 (4 parameter logistic fit, R² = 0.8049 and 0.8311 for LPS2336.HCl and ML335, respectively).

Finally, we evaluated the effect of the same 50 µM concentration of LPS2336.HCl in HEK cells expressing the human TREK2 clone. In these cells, only a mild increase in TREK-2 current was measured by patch clamp (2.78 folds, *p* = 0.0959, as compared to 10.77-fold in HEK-hTREK-1 cells).

### 3.4. Synthesis and Evaluation of LPS2336 Analogs

LPS2336 has been published as a synthesis intermediate for pyridazine acetylcholinesterase inhibitors together with several analogs [44]. We therefore decided to evaluate the ability of some of these analogs to activate TREK-1. Among the published analogs available, twelve were selected for their chemical diversity and were evaluated by thallium assay at 100 µM (Figure 5A,B). Interestingly, activity on TREK-1 was completely lost in every tested analog and the most active compound remained LPS2336.HCl. LPS6257, bearing a naphthalamide moiety, even produced a decrease in ΔFmax/F0 (−70% at 100 µM compared to the vehicle condition), which could reflect TREK-1 inhibition as well as cellular toxicity. With the aim of optimizing the bioactivity of LPS2336, we embarked on a structure–activity relationship study with the synthesis of 11 piperidine-based analogs 2a–k (Figure 5C). Analogs 2a–h, 2k were prepared by N-alkylation of the corresponding piperidine 1a-c with the appropriate halogenated alkyl under basic conditions (K_2_CO_3_ or DIPEA). We first prepared two simplified piperidine analogs where the benzyl moiety of LPS2336 was replaced by a hydrogen (2d) or a methyl (2b), which resulted in the loss of TREK-1 activation. N-alkylation of the benzylated piperidine platform with various hydrophobic chains (2a, 2c, 2e, 2f, 2g) led to a profound decrease in TREK-1 activation. The replacement of the nitrile function of LPS2336 by an ester (2h), acid (2i, obtained by saponification of 2h), amine (2j, obtained by reduction with LiAlH_4_ of LPS2336) and amide (2k) also severely impacted TREK-1 activation ability. We next turned our attention to piperazine analogs of LPS2336 (Figure 5C). We prepared 10 analogs in a two-step N,N’-dialkylation of piperazine through the addition of the appropriate halogenated moiety and obtained five N-benzylpiperazines (3a, 3aa, 4a, 4e, 4f), six trifluoromethyl-substituted benzylpiperazines (monosubstituted 3b–d, disubstituted 4b–d). Unfortunately, none of the piperidine and piperazine analogs were able to activate TREK-1. LPS2336, the compound originally identified by HTS, remains by far the best TREK-1 activator among all tested analogs.

### 3.5. Effect of LPS2336 on Pain Thresholds

Given the accumulating data in favor of the potential of TREK-1 pharmacological activation for pain relief [10,11,26,27,28,46], we tested whether LPS2336.HCl could reduce pain perception in mice. We used a model of paw subacute inflammation induced by carrageenan injection in CD1 mice and evaluated heat pain sensitivity using the Hargreaves plantar test (Figure 6). Following carrageenan injection, the pain threshold of animals decreases strongly, reflecting inflammatory pain hypersensitivity. When injected systemically, via intraperitoneal route, LPS2336 induced an increase in pain thresholds from 20 mg/kg (93.3 µmol/kg), similar to that induced by 3 mg/kg morphine (10.5 µmol/kg, Figure 6A). At 30 and 40 mg/kg, the effect of LPS2336 was only briefly increased at peak analgesia (15 min after injection) which did not translate into an overall increased analgesic effect when compared to the effect of the 20 mg/kg dose, as shown by the area under the curves.

Using the same model, we assessed the effect of central administrations of LPS2336. Intrathecal administration of LPS2336 was not effective in relieving pain in carrageenan-injected mice (Figure 6B). Only a slight, non-significant effect of LPS2336 on pain thresholds can be observed at the highest dose of 300 µg (1400 nmol) while 3 µg morphine (10.5 nmol) provided long-lasting analgesia. When injected via intracerebroventricular route, LPS2336 was only able to reverse the pain phenotype of mice at 100 µg (466.6 nmol), producing an effect similar to that of 1 µg morphine (3.5 nmol, Figure 6C). Systemic administration of LPS2336 is therefore a much more effective route to provide pain relief in this inflammatory pain model; although, a supraspinal effect of LPS2336 is observed at a very high dose.

### 3.6. Evaluation of the Adverse Effects Induced by LPS2336

To search for possible adverse effects induced by LPS2336 in vivo, we submitted CD1 mice to different confounding tests aiming at characterizing motor coordination, balance, muscle strength and spontaneous locomotor activity. Animals received an intraperitoneal injection of LPS2336.HCl at a dose of 10 to 50 mg/kg 15 min before the test, corresponding to the peak of analgesic effect previously observed. Animals performed normally when tasked to walk on a cylinder rotating at an increasing speed (rotarod test), whatever the dose administered (Figure 7A), showing normal motor coordination. On the horizontal bar challenge, animals receiving the highest dose of 50 mg/kg LPS2336 took longer to climb on a horizontal bar after grabbing it with their forepaws (horizontal bar test, Figure 7B). To assess the contribution of TREK-1 to this adverse effect, we submitted TREK-1 knock-out mice (which are of the C57Bl6/J background) to the same test, 15 min following the injection of 50 mg/kg LPS2336. First, we observed that WT littermates showed the same increase in time to climb than CD1 mice (Figure 7C). Second, there was no difference between WT and TREK-1 KO mice receiving LPS2336, ruling out a contribution of TREK-1 to this effect. We finally submitted the mice to a third test consisting in observing their spontaneous locomotion in an open arena (open field test). In this test, LPS2336 administration resulted in a decreased spontaneous locomotion from 30 mg/kg (Figure 7D). Again, while C57Bl6/J mice experienced the same adverse effect as CD1 mice at 50 mg/kg LPS2336, there was no difference between WT and TREK-1 KO mice (Figure 7E). Finally, we monitored the body temperature of mice for 45 min after intraperitoneal injection of LPS2336 at 40 mg/kg and could observe hypothermia in both WT and TREK-1 KO (−2.4 and −2.6 °C, respectively) following LPS2336 treatment (Figure 7F,G).

## 4. Discussion

This work shows that the thallium assay is adapted for the identification of potent TREK-1 activators among compounds collections, as it was used before to identify activators of Kir [47], TASK3 [32] and TREK2 channels [31,33]. Depending on the target and on the cellular expression model, appropriate optimization of the assay is essential. We provide insights for the optimization of the assay’s key conditions (thallium and DMSO concentrations) as well as data analysis (assay kinetics and hits detection). Optimal thallium concentration is particularly dependent on the nature (Tl^+^ conductance) and the expression level of the target. Indeed, we had to adapt thallium concentration between the HTS and the characterization of analogs, which was performed months later on a different batch of the same monoclonal cell line. We highly recommend running dose–response experiments to determine the optimal thallium concentration for any given experimental configuration. We also show that DMSO concentrations as low as 1–2% may strongly impact the assay. This is particularly important as DMSO is the standard solvent of chemical collections. Maximal tolerance of the assay to DMSO should therefore be determined systemically and will influence the concentration at which compounds can be screened.

Among the 1040 compounds tested, quinazoline RN-1-025 reached the positivity threshold once out of three replicates, and piperidine LPS2336 reached the positivity threshold three times with a higher Z score. The activity of both compounds could be confirmed by patch clamp, showing that our HTS conditions and hit detection threshold favor specificity over sensitivity (i.e., less false positive detections at the expense of possible false negatives). This relatively small-scale optimization provides a proof of principle for the feasibility of identifying TREK-1 activators. The proposed parameters could help scale up this assay to screen large compound collections. This approach may ultimately allow the identification of a set of TREK-1 activators with diverse properties (affinity, selectivity, solubility, off-targets, distribution and metabolism), providing a solid starting point for structure–activity relationship studies.

Since piperidine LPS2336 showed good solubility, particularly after the generation of a hydrochloride salt, and as it produces the most potent TREK-1 activation with an EC_50_ of 11.76 µM, we focused on this compound for optimization and further characterization. First, twelve existing analogs, and 21 newly synthetized analogs were screened. Interestingly, all of them showed little to no ability to activate TREK-1. We then tested the analgesic activity of LPS2336 in vivo and observed a plateau effect at 20 mg/kg, where the effect of LPS2336 was comparable to that of 3 mg/kg morphine. Although no deficit in motor coordination was observed with the rotarod test with doses up to 50 mg/kg, spontaneous locomotion was reduced from 30 mg/kg as seen with the open field test. This adverse effect was not different in TREK-1 KO mice, showing that it is not a consequence of TREK-1 activation. A similar observation was made regarding the decrease in body temperature observed in WT and KO mice at 40 mg/kg. While we have no means of identifying the off-targets involved in this effect, which could be unrelated to TREK-1, it is worth noting that most K2P channels are thermosensors [48]. A study recently reported an increase in body temperature in ovariectomized rats given ostruthin, a TREK-1/2 channel activator. The TASK-1 and TRAAK channels have also been identified in the preoptic area, which is important for thermoregulation [49]. Metabolic changes could also be at play. For example, TASK1 has previously been involved in the control of the thermogenic activity of brown adipose tissue [50]. These results reinforce the need for the discovery of more diverse TREK-1-activating compounds, which could come from the screening of larger compounds collections, and the need for systematic investigation of adverse effects in preclinical drug development studies. Using confounding tests, such as the open field test to look for potential adverse effects, is an easy and sensitive way to detect acute toxicity.

Finally, as it is yet unclear whether central or peripheral TREK-1 activation is important for analgesia, we performed central administrations of LPS2336. Intracerebroventricular injection of LPS2336 was poorly effective in raising pain thresholds in mice, with an analgesic effect only observed at the very high dose of 100 µg (466.6 nmol) while 1 µg morphine (3.5 nmol) is analgesic in this route of administration. When injected intrathecally, LPS2336 failed to exert any analgesic effect, even when used at 100 times the effective dose of morphine (133 times in molarity). Furthermore, when injected systemically, LPS2336 at 20 mg/kg (93.3 µmol/kg) was as effective as 3 mg/kg (10.5 µmol/kg) morphine, showing that peripheral TREK-1 channels might be much more important analgesic targets. This is particularly interesting when considering the risk of adverse effect linked to centrally acting drugs.

## 5. Conclusions

We show that thallium-based high throughput screening is a suitable way to identify new TREK-1 activators, even in small chemical collections. In our case, we identified two hits out of 1040 tested compounds whose activities could be confirmed using patch clamp. The risk of experimental and analytic bias is important with this type of assay, and we provide key optimization and control steps that should be taken into consideration when designing a screening project.

We also emphasize the importance of evaluating adverse effects (in WT and KO animals when possible) early in the development of compounds. The behavioral tests we compare in this study are quick and easy to perform for anyone with basic training in mice manipulation. We show that LPS2336 has a narrow therapeutic window, with an analgesic effect reaching a plateau at 20 mg/kg while a decreased spontaneous locomotor activity could be detected from 30 mg/kg (although no deficit in motor coordination could be observed at 50 mg/kg, the highest dose tested). Adverse effects were similar in TREK-1 KO mice, showing that these were mediated by off-targets. We finally show that LPS2336 analgesic activity is likely mediated at the periphery, as the compound is poorly effective when injected into the brain and spinal cord, as opposed to systemic delivery. Designing a peripherally restricted TREK-1 activator might therefore be an interesting strategy to benefit from the peripheral role of TREK-1 channels while minimizing the risk of centrally mediated adverse effects.

## Figures and Tables

**Figure 1 biomolecules-15-00740-f001:**
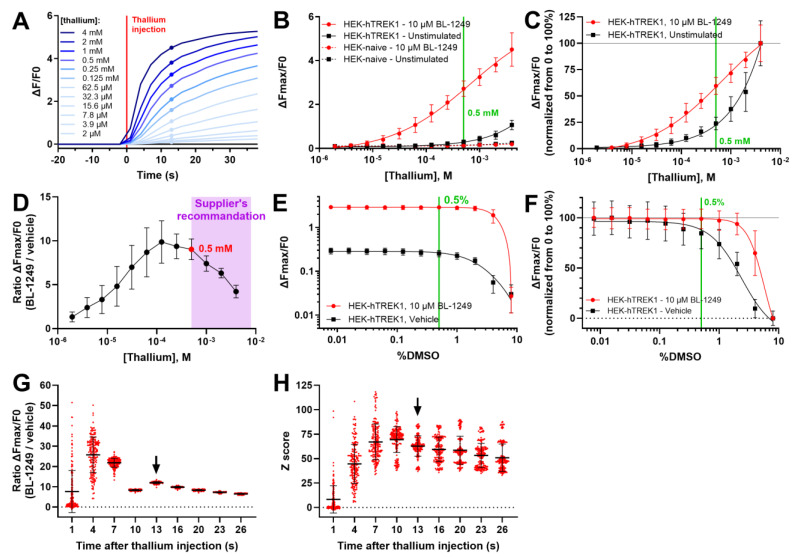
Optimization of the thallium assay. (**A–D**) Optimization of thallium concentration in naive and HEK-hTREK-1 cells treated with vehicle or 10 µM BL-1249. *n* = 42 wells over 3 plates. (**A**) Average normalized fluorescence traces of BL-1249-treated HEK-hTREK-1 cells. Dots represent the time-point used for subsequent ΔFmax/F0 extraction. (**B**) Thallium dose–effect curves constructed with ΔFmax/F0 from naive and HEK-hTREK-1 cells treated with BL-1249 or vehicle. Optimal thallium concentration, determined in D, is shown in green. (**C**) Normalization from 0 to 100% of the HEK-hTREK-1 curves from B. (**D**) In HEK-hTREK-1 cells, ΔFmax/F0 of BL-1249-treated samples was divided by the average ΔFmax/F0 of the vehicle group to evaluate the influence of thallium concentration on the separation between treated and non-treated cells. The purple zone defines the range of thallium concentrations recommended by the manufacturer. In red is the optimal thallium concentration used for subsequent experiments and high throughput screening. (**E**,**F**) Optimization of DMSO concentration in naive and HEK-hTREK-1 cells treated with vehicle or 10 µM BL-1249. *n* = 48 wells over 3 plates. Normalized DMSO dose–effect curves are shown before (**E**) and after (**F**) normalization from 0 to 100%. The final DMSO concentration used in high throughput screening is indicated in green. (**G**,**H**) Optimization of the time-point used for Fmax extraction in HEK-hTREK-1 cells treated with 10 µM BL-1249 or vehicle. *n* = 180 wells over 4 plates. (**G**) ΔFmax/F0 of BL-1249-treated samples were divided by the average ΔFmax/F0 of the vehicle group to determine which maximum time-point allows the best separation between treated and control wells. The arrow indicates the time-point retained for subsequent analyses and high throughput screening as the best compromise between separation and variability. (**H**) Z scores calculated in BL-1249 treated wells. In all panels, means ± standard deviations are shown.

**Figure 2 biomolecules-15-00740-f002:**
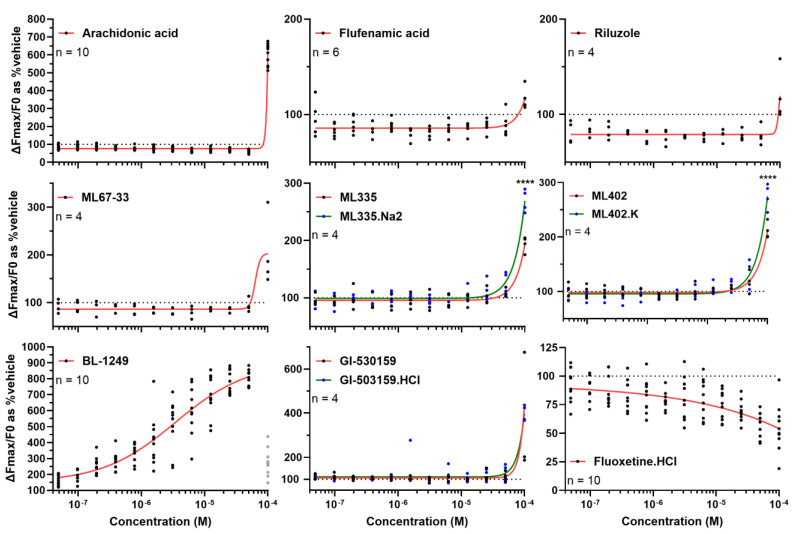
Dose–response curves of different known TREK-1 modulators. Dose–response curves were generated for 9 compounds in HEK-hTREK-1 cells using the previously optimized thallium assay conditions (0.5 mM thallium, 0.5% DMSO, Fmax measured up to 13 s after thallium injection). Some compounds were salified (ML335.Na2, ML402.K, GI-503159.HCl) to improve solubility, and the results obtained with salts are compared with those obtained with native molecules. For Bl-1249, the 100 µM datapoints (shown in gray) were excluded from the logistic regression (least square fit, 4 parameters) as visible toxicity occurred. Results are normalized to vehicle-treated wells. n numbers (wells) are indicated on each panel. Means ± standard deviations are shown. **** *p* < 0.0001 vs. salified compound, 2way ANOVA followed by Šídák’s multiple comparisons test.

**Figure 3 biomolecules-15-00740-f003:**
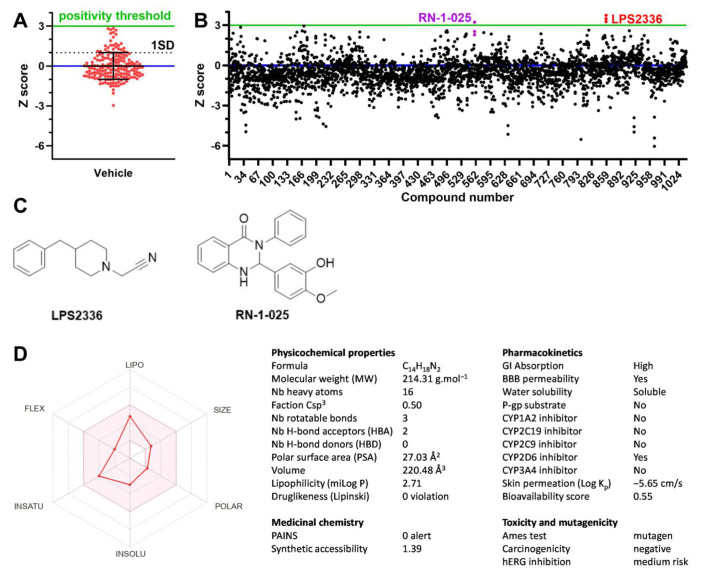
Screening for TREK-1 activators in a collection of 1040 compounds. Compounds from the Essential Chemical Library (ChemBioFrance) were screened at 25 µM on HEK-hTREK-1 cells, on four 384-well plates replicated 3 times. Previously determined optimal conditions were used (0.5 mM thallium, 0.5% DMSO, Fmax measured up to 13 s after thallium injection). (**A**) Z-score of 180 vehicle-treated wells (HEK-hTREK-1 cells) spread over the 12 screening plates. A positivity threshold of 3 standard deviations was determined for high throughput screening to minimize false positive detection. (**B**) High throughput screening results expressed as Z scores. 1040 triplicates are shown. One compound, LPS2336, was over the positivity threshold 3/3 times (triplicates in red). One compound, RN-1-025, was over the positivity threshold 1/3 time (triplicates in purple). (**C**) Structures of LPS2336 and RN-1-025. (**D**) In silico analysis of LPS2336. (**Left**) Oral bioavailability radar of piperidine LPS2336 (shown as red line). The pink area depicts a suitable physicochemical space for bioavailable drugs. INSATU: insaturation; POLAR: polarity; INSOLU: insolubility; LIPO: lipophilicity; FLEX: flexibility; SIZE: molecular weight. (**Right**) Summary table of LPS2336’s predicted physicochemical properties, pharmacokinetic profile, drug-likeness and toxicity by molinspiration, SwissADME and preADMET.

**Figure 4 biomolecules-15-00740-f004:**
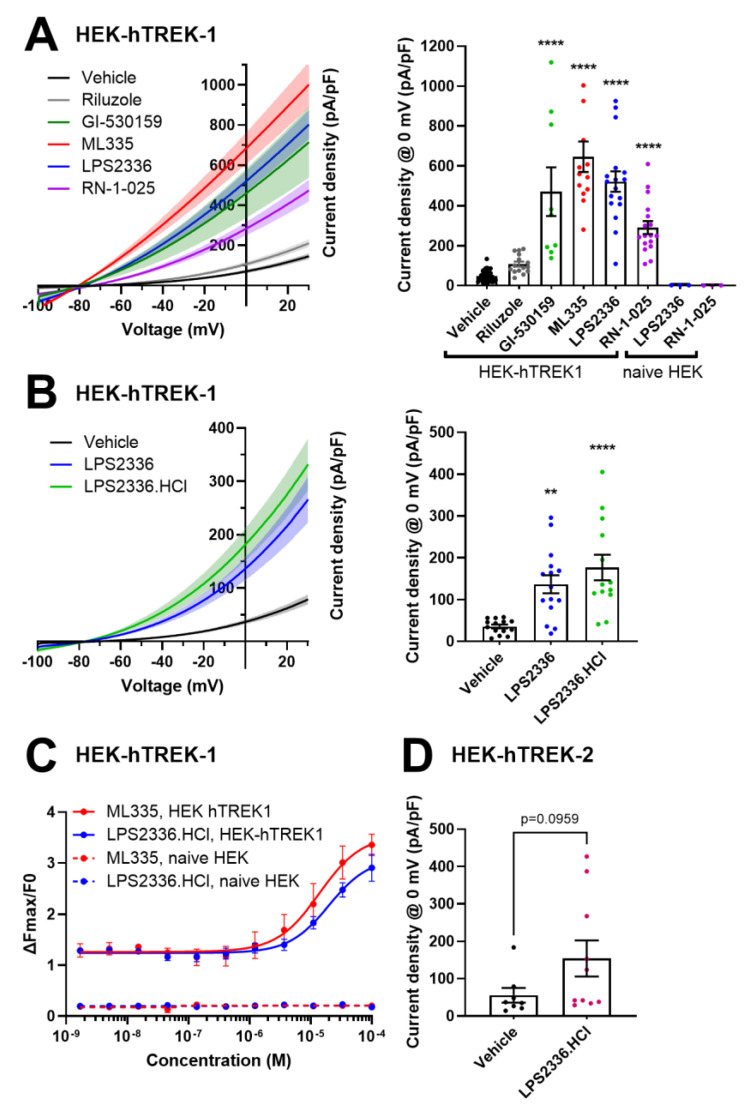
TREK-1 activation by LPS2336 and RN-1-025. (**A**,**B**) Current densities recorded in HEK-hTREK-1 cells treated with 50 µM riluzole (*n* = 18 cells), GI-530159 (*n* = 9), ML335 (*n* = 13), LPS2336 (*n* = 17), RN-1-025 (*n* = 17) and vehicle (0.01% DMSO, *n* = 41) (**A**), and LPS2336 (*n* = 15), salified LPS2336.HCl (*n* = 13) and vehicle (0.01% DMSO, *n* = 14) (**B**). Left panels show current-voltage relationships from −100 to +30 mV. Currents are normalized by cell capacitances. Right panels show average current densities ± SEM recorded at 0 mV. In (**A**) are also shown currents recordings made in untransfected HEK293 cells treated with LPS2336 or RN-1-025 (*n* = 3 cells). (**C**) Dose–response curves of LPS2336.HCl and ML335 generated by thallium assay on HEK-hTREK-1 and untransfected HEK293 cells (0.5 mM thallium, 0.5% DMSO, Fmax measured up to 13 s after thallium injection). *n* = 5 wells. Mean ΔFmax/F0 ± SEM are shown. (**D**) Average current densities ± SEM recorded at 0 mV in HEK-hTREK2 cells. ** *p* < 0.01, **** *p* < 0.0001 vs. vehicle, Kruskal–Wallis test followed by Dunn’s post hoc test (**A**), one way ANOVA followed by Dunnett’s post hoc test (**B**) and Mann–Whitney test (**D**).

**Figure 5 biomolecules-15-00740-f005:**
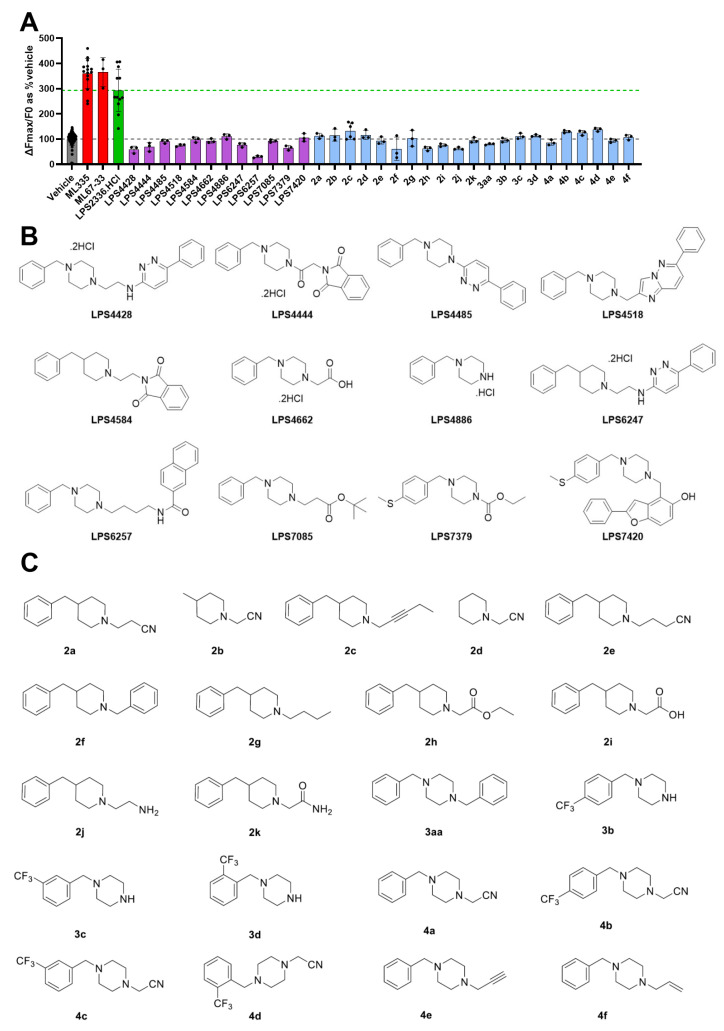
Evaluation of LPS2336 analogs. (**A**) 33 analogs of LPS2336 were tested on HEK-hTREK-1 cells using the thallium assay (2 mM thallium, 0.5% DMSO, Fmax measured up to 13 s after thallium injection). All compounds are used at 100 µM final. ML335 (*n* = 15), ML67-33 (*n* = 3) and LPS2336.HCl (*n* = 12 wells) were used as positive controls. *n* = 3 wells for all LPS2336 analogs but 2c (*n* = 6). Results are shown normalized to vehicle-treated wells (*n* = 240). Gray and green dashed lines indicate the mean of vehicle and LPS2336.HCl groups, respectively. Means ± standard deviations are shown. Structures of the 12 analogs from Contreras et al., (2001) [44] and of the 21 analogs synthetized in the frame of this study are shown in (**B**) and (**C**), respectively.

**Figure 6 biomolecules-15-00740-f006:**
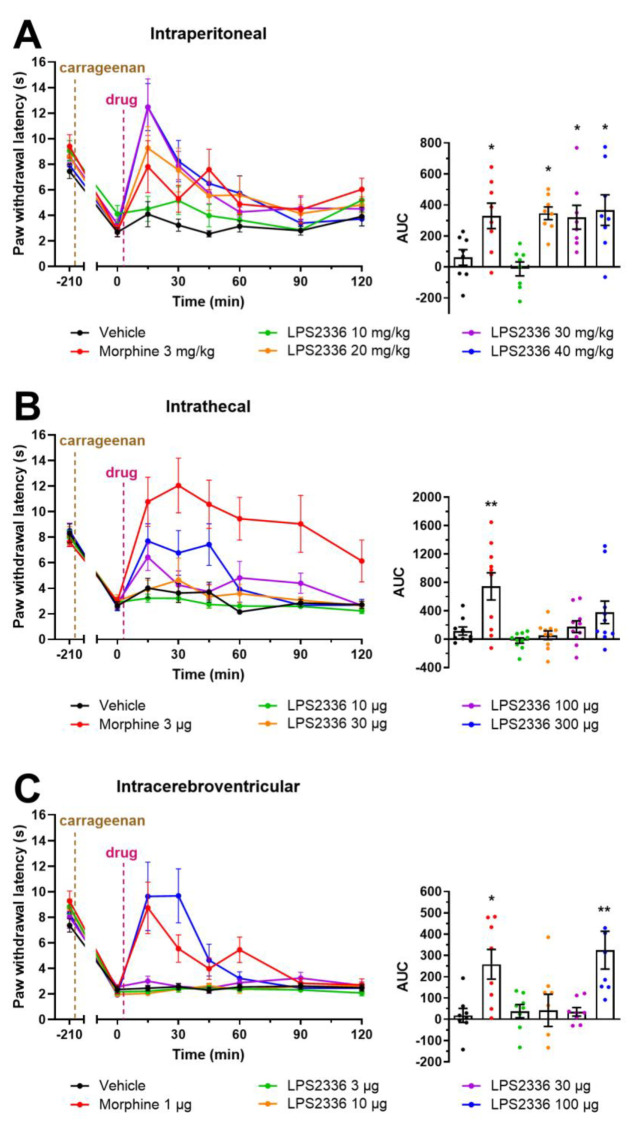
Effect of systemic and central treatment with LPS2336 on pain thresholds in a mouse model of inflammatory pain. Subacute paw inflammation was induced in WT CD1 mice by intraplantar injection of carrageenan (20 µL, 2%) 3.5 h before administration of different doses of LPS2336.HCl, morphine or vehicle (saline) given intraperitoneally (**A**), intrathecally (**B**) or intracerebroventricularly (**C**). Thermal pain thresholds were evaluated using the Hargreaves test before carrageenan injection (−210 min) and before and for 2 h after treatment (0–120 min). Doses given account for LPS2336 after removal of the HCl salt. Left panels show time course evaluation of pain thresholds and right panels show the area under curve measured from 0 to 120 min. Mean ± SEM are shown. *n* = 8 animals in all groups. * *p* < 0.05, ** *p* < 0.01 vs. vehicle, one way ANOVA followed by Dunnett’s post hoc test.

**Figure 7 biomolecules-15-00740-f007:**
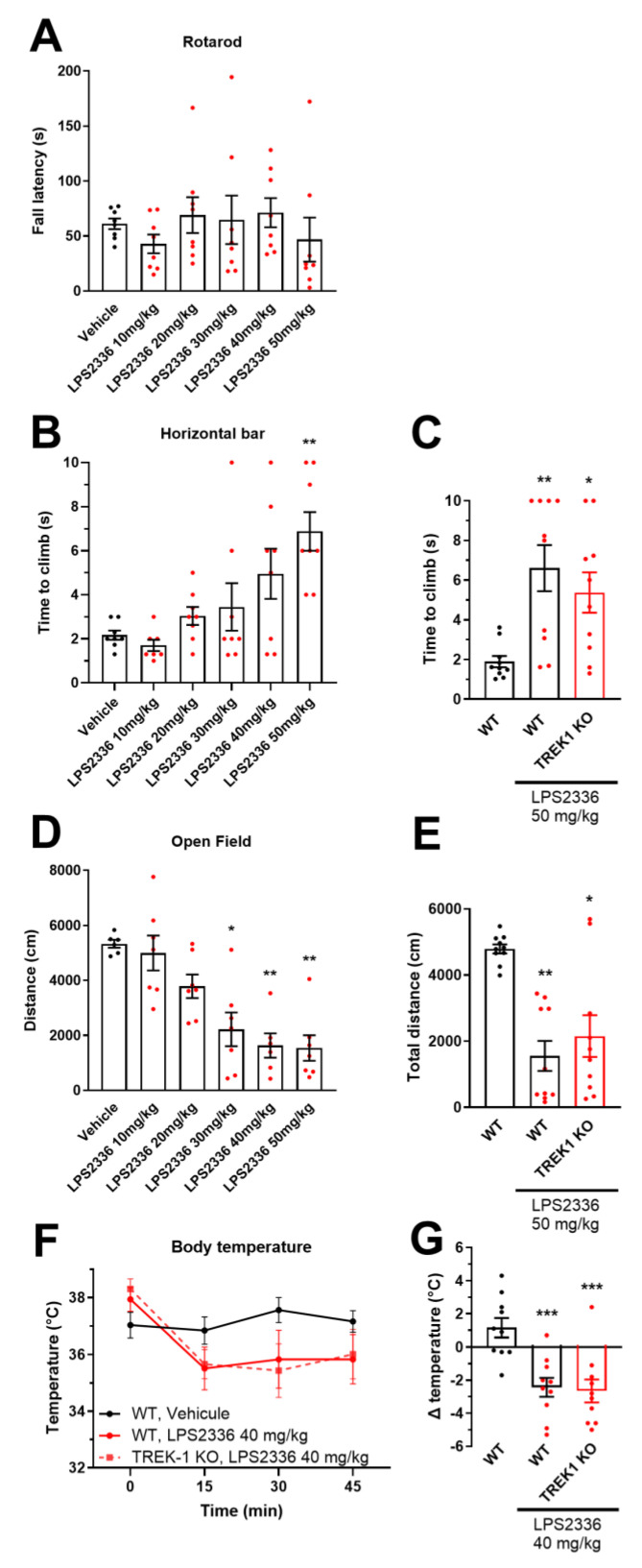
Adverse effects of systemic treatment with LPS2336. (**A**) Evaluation of motor coordination, balance and fatigue using the rotarod test in WT CD1 mice 15 min after intraperitoneal injection of LPS2336 at 0, 10, 20, 30, 40 and 50 mg/kg (*n* = 8 animals). (**B**) Evaluation of motor coordination and grip strength using the horizontal bar test in WT CD1 mice 15 min after intraperitoneal injection of LPS2336 at 0, 10, 20, 30, 40 and 50 mg/kg (*n* = 8). (**C**) The same horizontal bar test was performed in WT and TREK-1 KO littermates of C57Bl6/J background injected with 0 or 50 mg/kg LPS2336 (*n* = 10). (**D**) Evaluation of spontaneous locomotor activity of WT CD1 mice in an open field 15 min after intraperitoneal injection of LPS2336 at 0, 10, 20, 30, 40 and 50 mg/kg (*n* = 6–7). (**E**) The same open field test was performed in WT and TREK-1 KO littermates of C57Bl6/J background injected with 0 or 50 mg/kg LPS2336 (*n* = 10). (**F**) Rectal temperature measured in WT and TREK-1 KO littermates of C57Bl6/J background injected intraperitoneally with 0 or 40 mg/kg LPS2336, before and 15, 30 and 45 min after injection. (**G**) For each animal in (**F**), the body temperature before injection was subtracted from the body temperature 15 min after injection. * *p* < 0.05, ** *p* < 0.01, *** *p* < 0.001 vs. vehicle. Kruskall-Wallis followed by Dunn’s multiple comparisons test (**A**–**E**) and one way ANOVA followed by Dunett’s multiple comparisons test (**G**). Doses given account for LPS2336 after removal of the HCl salt.

## Data Availability

The raw data supporting the conclusions of this article will be made available by the authors on request.

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
