# Peer review of "LPS2336, a New TREK-1 Channel Activator Identified by High Throughput Screening"

_biomolecules, 2025, doi:10.3390/biom15050740_

Round 1
Reviewer 1 Report
Comments and Suggestions for Authors
The study aims to identify and characterize novel pharmacological activators of the TREK-1 potassium channel, with potential application in pain relief. Specifically, it reports the discovery of LPS2336 via high-throughput thallium flux screening, evaluates its analogs, and assesses both its in vitro activity and in vivo analgesic and adverse effects.
The topic seems to me quite original and relevant to current efforts in analgesic drug development. TREK-1 is a well-recognized but underexploited target for pain therapy, and the work addresses this gap in the field: i.e. the lack of potent, selective, and bioavailable TREK-1 activators. I think that this study by identifying LPS2336, contributes significantly to both pharmacological tool development and even in some degree to preclinical trials, since it explores efficacy and off-target activity of LPS2336.
In my opinion the study advances the field in several areas:
- Demonstrating a well-optimized thallium-based HTS assay for K2P channel activators.
- Identifying a novel TREK-1 activator, LPS2336, with ECâ‚…â‚€ ~11.8 µM.
- Providing SAR analysis through the synthesis and screening of 33 analogs.
- Performing in vivo analgesia assays that directly compare peripheral and central effects.
- Documenting off-target adverse effects and demonstrating that they are TREK-1-independent using knockout controls.
It seems to me that additional specificity controls, such as effects on other K2P subtypes (e.g., TREK-2, TASK-1), could strengthen the conclusion about TREK-1 selectivity.
I think the conclusions are well-supported by the data. First of all, LPS2336 is confirmed to activate TREK-1 in multiple assays (thallium flux, patch-clamp). Analog screening shows a narrow SAR suggesting posible selectivity of this compound. In adddition, behavioral assays and the use of TREK-1 KO mice convincingly demonstrate that analgesic effects are mediated via TREK-1, whereas sedative and hypothermic effects are off-target.
In my opinion, the manuscript is well-referenced and up-to-date, and cites key studies on TREK-1 function, prior tool compounds (e.g., ML335), and relevant behavioral models. The reference formatting should be checked for consistency (especially in the discussion), but the selection itself is appropriate.
Figures are clear, well-labeled, and informative. Figure 1 does an excellent job illustrating assay optimization. Behavioral results (e.g., Figure 6 and 7) are effectively presented with appropriate statistical annotations.
Author Response
Thank you very much for taking the time to evaluate our work. Please find our point-by-point response below. A new version of the manuscript is available with corrections in track changes, taking all 3 reviewers’ comments into consideration.
Comments 1:
The study aims to identify and characterize novel pharmacological activators of the TREK-1 potassium channel, with potential application in pain relief. Specifically, it reports the discovery of LPS2336 via high-throughput thallium flux screening, evaluates its analogs, and assesses both its in vitro activity and in vivo analgesic and adverse effects.
The topic seems to me quite original and relevant to current efforts in analgesic drug development. TREK-1 is a well-recognized but underexploited target for pain therapy, and the work addresses this gap in the field: i.e. the lack of potent, selective, and bioavailable TREK-1 activators. I think that this study by identifying LPS2336, contributes significantly to both pharmacological tool development and even in some degree to preclinical trials, since it explores efficacy and off-target activity of LPS2336.
In my opinion the study advances the field in several areas:
- Demonstrating a well-optimized thallium-based HTS assay for K2P channel activators.
- Identifying a novel TREK-1 activator, LPS2336, with ECâ‚…â‚€ ~11.8 µM.
- Providing SAR analysis through the synthesis and screening of 33 analogs.
- Performing in vivo analgesia assays that directly compare peripheral and central effects.
- Documenting off-target adverse effects and demonstrating that they are TREK-1-independent using knockout controls.
Response 1:
We are thankful for this positive assessment of the quality of our work.
Comments 2:
It seems to me that additional specificity controls, such as effects on other K2P subtypes (e.g., TREK-2, TASK-1), could strengthen the conclusion about TREK-1 selectivity.
Response 2:
As suggested by you and another reviewer, we have tested the effect of LPS2336 on TREK-2, the most closely-related channel to TREK-1. We have measured the TREK-2 current produced in HEK cells upon depolarization after exposure to 50 µM LPS2336, similarly to what we have done for TREK-1. We could see a mild increase in TREK-2 current (2.78 folds, p=0.0959), which is smaller than the increase observed with TREK-1 currents (10.77 fold). This data is now shown in Figure 4D and presented in the manuscript p13.
Comments 3:
I think the conclusions are well-supported by the data. First of all, LPS2336 is confirmed to activate TREK-1 in multiple assays (thallium flux, patch-clamp). Analog screening shows a narrow SAR suggesting posible selectivity of this compound. In adddition, behavioral assays and the use of TREK-1 KO mice convincingly demonstrate that analgesic effects are mediated via TREK-1, whereas sedative and hypothermic effects are off-target.
In my opinion, the manuscript is well-referenced and up-to-date, and cites key studies on TREK-1 function, prior tool compounds (e.g., ML335), and relevant behavioral models. The reference formatting should be checked for consistency (especially in the discussion), but the selection itself is appropriate.
Figures are clear, well-labeled, and informative. Figure 1 does an excellent job illustrating assay optimization. Behavioral results (e.g., Figure 6 and 7) are effectively presented with appropriate statistical annotations.
Response 3:
We again thank you for these positive comments. We indeed had a Zotero problem with a few references and we apologies for that. We have updated the references which are all properly formatted for Biomolecules now.
Reviewer 2 Report
Comments and Suggestions for Authors
This manuscript describes the optimization of a high-throughput screening (HTS) method to discover and characterize LPS2336, a novel activator of the TREK-1 potassium channel. Using an HTS technique based on thallium flux monitoring, the authors screened 1040 compounds from the French National Chemical Library and identified LPS2336 as a potent TREK-1 activator with an EC50 of 11.76 μM. Among the 33 LPS2336 analogs tested, none maintained activity on TREK-1. In in vivo experiments, LPS2336 demonstrated antinociceptive effects when administered systemically, showed limited effects with intracerebroventricular administration, but no effect with intrathecal administration. These findings suggest the importance of peripheral TREK-1 channels in pain relief. Additionally, LPS2336 induced sedation and hypothermia at doses close to its analgesic effective dose. These adverse effects were also observed in TREK-1 knockout mice, confirming they are mediated by off-target mechanisms.The manuscript is methodologically sound with well-executed experiments. The detailed description of the optimization process for the thallium-based HTS method provides valuable information for future research. The comparison of drug effects through various administration routes and the evaluation of adverse effects using TREK-1 knockout mice are strengths of the study.
However, there are significant limitations regarding the characterization of drug selectivity. The following detailed comments address these and other concerns that should be addressed.
- Redundancy between Abstract and Introduction:
There is significant overlap between the Abstract and the final section of the Introduction, with several sentences appearing nearly identical. The Introduction should focus more on providing research context, background literature, and clear motivation for the study. I recommend restructuring the final paragraphs of the Introduction to emphasize the scientific rationale behind the study design rather than repeating the methodological approach and findings that are already summarized in the Abstract. - Insufficient Selectivity Testing:
The manuscript focuses exclusively on TREK-1 screening without providing essential selectivity data across other K2P channels. This raises questions about the specificity of LPS2336. While comprehensive screening across all K2P channels would be ideal, at minimum, selectivity testing should be performed against structurally similar subtypes such as TREK-2 and TRAAK using either patch clamp or thallium assay methods. This is particularly relevant given the authors' conclusion regarding peripheral effects, as nociceptors express multiple K2P channels including TREK, TRAAK, and TRESK. Determining which channels are actually targeted would significantly strengthen the pharmacological characterization of LPS2336. - Need for Functional Neuronal Studies:
Electrophysiological experiments on small-sized DRG neurons would provide valuable insight into the compound's mechanism of action. Specifically, measuring changes in rheobase current and effects on voltage-operated Na+ and K+ channels would help explain how LPS2336 influences action potential firing. This is particularly important given the observation that mice administered 50mg/kg LPS2336 took longer to climb the horizontal bar, suggesting possible effects on motor neurons. Without such experiments, it remains unclear whether the observed effects result from non-selective K2P channel activation increasing rheobase or from modulation of voltage-operated channels. - Limited Discussion of Adverse Effects:
The Discussion section requires more comprehensive analysis of the observed hypothermia and reduced motor activity. These effects suggest that LPS2336 might activate multiple K2P family members, potentially influencing temperature-regulating centers in the hypothalamus. The authors should discuss relevant literature on K2P channel expression in hypothalamic nuclei and their role in temperature regulation. Additionally, addressing whether there are known connections between K2P channel modulation and thermoregulation would enhance the mechanistic understanding of the observed adverse effects. - Suggestion for In Silico ADME Analysis:
Despite the narrow therapeutic window of LPS2336, it could serve as a backbone for future lead compound development. To facilitate this process, I recommend including in silico ADME (Absorption, Distribution, Metabolism, Excretion) analysis using tools such as SwissADME. Presenting these results as a figure with accompanying interpretation would provide valuable information about potential toxicity issues and pharmacokinetic properties. This would benefit researchers seeking to develop derivatives with improved safety profiles while maintaining efficacy.
Author Response
Thank you very much for taking the time to evaluate our work. Please find our point-by-point response below. A new version of the manuscript is available with corrections in track changes, taking all 3 reviewers’ comments into consideration.
Comments 1:
This manuscript describes the optimization of a high-throughput screening (HTS) method to discover and characterize LPS2336, a novel activator of the TREK-1 potassium channel. Using an HTS technique based on thallium flux monitoring, the authors screened 1040 compounds from the French National Chemical Library and identified LPS2336 as a potent TREK-1 activator with an EC50 of 11.76 μM. Among the 33 LPS2336 analogs tested, none maintained activity on TREK-1. In in vivo experiments, LPS2336 demonstrated antinociceptive effects when administered systemically, showed limited effects with intracerebroventricular administration, but no effect with intrathecal administration. These findings suggest the importance of peripheral TREK-1 channels in pain relief. Additionally, LPS2336 induced sedation and hypothermia at doses close to its analgesic effective dose. These adverse effects were also observed in TREK-1 knockout mice, confirming they are mediated by off-target mechanisms.The manuscript is methodologically sound with well-executed experiments. The detailed description of the optimization process for the thallium-based HTS method provides valuable information for future research. The comparison of drug effects through various administration routes and the evaluation of adverse effects using TREK-1 knockout mice are strengths of the study.
However, there are significant limitations regarding the characterization of drug selectivity. The following detailed comments address these and other concerns that should be addressed.
- Redundancy between Abstract and Introduction:
There is significant overlap between the Abstract and the final section of the Introduction, with several sentences appearing nearly identical. The Introduction should focus more on providing research context, background literature, and clear motivation for the study. I recommend restructuring the final paragraphs of the Introduction to emphasize the scientific rationale behind the study design rather than repeating the methodological approach and findings that are already summarized in the Abstract.
Response 1:
Thank you for this thorough review of our work. We rewrote the final part of the introduction (p2), focusing on the challenges of screening K2P channels activators. We also emphasize the importance of early adverse effects evaluation in the preclinical setting. The scientific rationale and background are detailed in the first half of the introduction.
Comments 2:
Insufficient Selectivity Testing:
The manuscript focuses exclusively on TREK-1 screening without providing essential selectivity data across other K2P channels. This raises questions about the specificity of LPS2336. While comprehensive screening across all K2P channels would be ideal, at minimum, selectivity testing should be performed against structurally similar subtypes such as TREK-2 and TRAAK using either patch clamp or thallium assay methods. This is particularly relevant given the authors' conclusion regarding peripheral effects, as nociceptors express multiple K2P channels including TREK, TRAAK, and TRESK. Determining which channels are actually targeted would significantly strengthen the pharmacological characterization of LPS2336.
Response 2:
As suggested by you and another reviewer, we have tested the effect of LPS2336 on TREK-2, the most closely-related channel to TREK-1. We have measured the TREK-2 current produced in HEK cells upon depolarization after exposure to 50 µM LPS2336, similarly to what we have done for TREK-1. We could see a mild increase in TREK-2 current (2.78 folds, p=0.0959), which is smaller than the increase observed with TREK-1 currents (10.77 fold). This data is now shown in Figure 4D and presented in the manuscript p13.
Comments 3:
Need for Functional Neuronal Studies:
Electrophysiological experiments on small-sized DRG neurons would provide valuable insight into the compound's mechanism of action. Specifically, measuring changes in rheobase current and effects on voltage-operated Na+ and K+ channels would help explain how LPS2336 influences action potential firing. This is particularly important given the observation that mice administered 50mg/kg LPS2336 took longer to climb the horizontal bar, suggesting possible effects on motor neurons. Without such experiments, it remains unclear whether the observed effects result from non-selective K2P channel activation increasing rheobase or from modulation of voltage-operated channels.
Response 3:
Knowing the effect of LPS2336 (or any TREK-1 activator) on DRG neurons excitability would be very valuable information. However, we found that about 5% of DRG neurons express TREK-1, which makes such an experiment complicated. Furthermore, TREK-1 positive neurons cannot be identified on the basis of leak potassium currents as we lack pharmacological tools to isolate TREK-1 currents from other K2P currents. We are currently working on developing a reporter mouse that would allow such work for later studies.
Comments 4:
Limited Discussion of Adverse Effects:
The Discussion section requires more comprehensive analysis of the observed hypothermia and reduced motor activity. These effects suggest that LPS2336 might activate multiple K2P family members, potentially influencing temperature-regulating centers in the hypothalamus. The authors should discuss relevant literature on K2P channel expression in hypothalamic nuclei and their role in temperature regulation. Additionally, addressing whether there are known connections between K2P channel modulation and thermoregulation would enhance the mechanistic understanding of the observed adverse effects.
Response 4:
Although the observed adverse reactions might very well be due to other K2P channels, many mechanisms not involving K2P channels might also be at play, and we have no way of precisely identifying the source of these off-target effects. Nevertheless, we enriched the discussion with elements regarding the role of K2P channels in thermoregulation (p20).
Comments 5:
Suggestion for In Silico ADME Analysis:
Despite the narrow therapeutic window of LPS2336, it could serve as a backbone for future lead compound development. To facilitate this process, I recommend including in silico ADME (Absorption, Distribution, Metabolism, Excretion) analysis using tools such as SwissADME. Presenting these results as a figure with accompanying interpretation would provide valuable information about potential toxicity issues and pharmacokinetic properties. This would benefit researchers seeking to develop derivatives with improved safety profiles while maintaining efficacy.
Response 5:
Thank you for this suggestion. We agree that in silico profiling provides a valuable preliminary evaluation of pharmacokinetic and safety parameters. We have performed in silico analysis of LPS2336 using molinspiration, SwissADME and preADMET and included the predicted properties in Figure 3 and in the results p11. We also added a brief interpretation of the key parameters.
Reviewer 3 Report
Comments and Suggestions for Authors
The manuscript entitled “LPS2336, a new TREK-1 channel activator identified by high-throughput screening” is highly relevant to pain research and makes a valuable contribution to advancing our understanding of TREK-1 modulation as a therapeutic target.
I have a few questions for the authors:
-
Did you apply bioisosteric replacement strategies to the LPS2336 molecule in an effort to improve selectivity, permeability, or reduce adverse effects?
-
Have you considered comparing the equipotency of LPS2336 and morphine by converting the doses to micromoles per kilogram (µmol/kg) for a more accurate comparison of pharmacological potency?
-
Do you plan to conduct chronic administration studies to evaluate the potential for tolerance development or cumulative effects over time?
The authors might discuss the structural limitation of the molecule compared to inactive analogues and include a brief mention of future screening or optimization strategies.
Author Response
Thank you very much for taking the time to evaluate our work. Please find our point-by-point response below. A new version of the manuscript is available with corrections in track changes, taking all 3 reviewers’ comments into consideration.
Comments 1:
The manuscript entitled “LPS2336, a new TREK-1 channel activator identified by high-throughput screening” is highly relevant to pain research and makes a valuable contribution to advancing our understanding of TREK-1 modulation as a therapeutic target.
I have a few questions for the authors:
Did you apply bioisosteric replacement strategies to the LPS2336 molecule in an effort to improve selectivity, permeability, or reduce adverse effects?
Response 1:
To respond to a question asked by reviewer 2, we included in silico ADMET prediction of LPS2336 (Figure 3 and results p11). SAR study aimed to explore the chemical space around this hit and bioisosteric replacement to identify alternative moieties to replace the cyanide (CN) and improve the water solubility (piperazine).
Comments 2:
Have you considered comparing the equipotency of LPS2336 and morphine by converting the doses to micromoles per kilogram (µmol/kg) for a more accurate comparison of pharmacological potency?
Response 2:
It is true that we should have provided doses in moles to accurately compare the effect of morphine and LPS2336. We have added this information to the manuscript in the results p15 and in the discussion p20.
Comments 3:
Do you plan to conduct chronic administration studies to evaluate the potential for tolerance development or cumulative effects over time?
Response 3:
This would have been the next step in the development of LPS2336, as we have previously done for REN28, another TREK-1 activator (Busserolles et al., 2019, https://doi.org/10.1111/bph.15243). However, due to the adverse effects we report with this molecule, we won’t conduct further preclinical studies on LPS2336 and rather focus on identifying new compounds with reduced off-target effects.
Comments 4:
The authors might discuss the structural limitation of the molecule compared to inactive analogues and include a brief mention of future screening or optimization strategies.
Response 4:
In silico ADMET prediction of LPS2336 suggests it is a good starting point but requires structural refinement. About the future screening optimization strategies, we have added the following paragraph in the discussion p19:
This relatively small scale optimization provides a proof of principle for the feasibility of identifying TREK-1 activators. The proposed parameters could help scaling up this assay to screen large compounds collections. This approach may ultimately allow the identification of a set of TREK-1 activators with diverse properties (affinity, selectivity, solubility, off-targets, distribution and metabolism) providing a solid starting point for struc-ture-activity relationship studies.
Round 2
Reviewer 2 Report
Comments and Suggestions for Authors
I have reviewed the revised manuscript and the authors' response letter. I am satisfied that the authors have adequately addressed all the questions and concerns raised during the review process.
I have no more comments and questions.